# The Current Status of Research on High-Density Lipoproteins (HDL): A Paradigm Shift from HDL Quantity to HDL Quality and HDL Functionality

**DOI:** 10.3390/ijms23073967

**Published:** 2022-04-02

**Authors:** Kyung-Hyun Cho

**Affiliations:** 1LipoLab, Yeungnam University, Gyeongsan 38541, Korea; chok@yu.ac.kr; Tel.: +82-53-964-1990; Fax: +82-53-965-1992; 2Raydel Research Institute, Medical Innovation Complex, Daegu 41061, Korea

**Keywords:** high-density lipoprotein (HDL), low-density lipoproteins (LDL), cholesterol, HDL quantity, HDL quality, HDL functionality

## Abstract

The quantity of high-density lipoproteins (HDL) is represented as the serum HDL-C concentration (mg/dL), while the HDL quality manifests as the diverse features of protein and lipid content, extent of oxidation, and extent of glycation. The HDL functionality represents several performance metrics of HDL, such as antioxidant, anti-inflammatory, and cholesterol efflux activities. The quantity and quality of HDL can change during one’s lifetime, depending on infection, disease, and lifestyle, such as dietary habits, exercise, and smoking. The quantity of HDL can change according to age and gender, such as puberty, middle-aged symptoms, climacteric, and the menopause. HDL-C can decrease during disease states, such as acute infection, chronic inflammation, and autoimmune disease, while it can be increased by regular aerobic exercise and healthy food consumption. Generally, high HDL-C at the normal level is associated with good HDL quality and functionality. Nevertheless, high HDL quantity is not always accompanied by good HDL quality or functionality. The HDL quality concerns the morphology of the HDL, such as particle size, shape, and number. The HDL quality also depends on the composition of the HDL, such as apolipoproteins (apoA-I, apoA-II, apoC-III, serum amyloid A, and α-synuclein), cholesterol, and triglyceride. The HDL quality is also associated with the extent of HDL modification, such as glycation and oxidation, resulting in the multimerization of apoA-I, and the aggregation leads to amyloidogenesis. The HDL quality frequently determines the HDL functionality, which depends on the attached antioxidant enzyme activity, such as the paraoxonase and cholesterol efflux activity. Conventional HDL functionality is regression, the removal of cholesterol from atherosclerotic lesions, and the removal of oxidized species in low-density lipoproteins (LDL). Recently, HDL functionality was reported to expand the removal of β-amyloid plaque and inhibit α-synuclein aggregation in the brain to attenuate Alzheimer’s disease and Parkinson’s disease, respectively. More recently, HDL functionality has been associated with the susceptibility and recovery ability of coronavirus disease 2019 (COVID-19) by inhibiting the activity of severe acute respiratory syndrome coronavirus 2 (SARS-CoV-2). The appearance of dysfunctional HDL is frequently associated with many acute infectious diseases and chronic aging-related diseases. An HDL can be a suitable biomarker to diagnose many diseases and their progression by monitoring the changes in its quantity and quality in terms of the antioxidant and anti-inflammatory abilities. An HDL can be a protein drug used for the removal of plaque and as a delivery vehicle for non-soluble drugs and genes. A dysfunctional HDL has poor HDL quality, such as a lower apoA-I content, lower antioxidant ability, smaller size, and ambiguous shape. The current review analyzes the recent advances in HDL quantity, quality, and functionality, depending on the health and disease state during one’s lifetime.

## 1. Introduction: What Are HDL Quantities, Qualities, and Functionalities?

The general factors that are considered in terms of HDL quantity are the serum HDL-C level and the amount of cholesterol in the HDL, which can be detected by enzymatic determination. HDL quantity is expressed simply as a number alongside an mg/dL or mmol unit. In contrast, HDL quality is reflected by more diverse features of the particle morphology because HDL is a highly complex structure [1], consisting of many lipids (cholesterol, triglyceride, and phospholipid) and proteins (apolipoproteins and enzymes). HDL quality is related to the morphology of HDL particles, such as the shape, size, and composition of lipids and proteins in the particles. HDL quality also includes the extent of the oxidation and glycation of the components in HDL. The HDL functionality is represented as the antioxidant ability and cholesterol efflux activity for the prevention of LDL oxidation and the regression of atherosclerotic plaque, respectively [2].

The quantity and quality of HDL can influence the overall health status of blood vessels and the properties of LDL, such as an oxidized LDL (oxLDL), which is considered a major culprit of cardiovascular disease (CVD) to facilitate the growth of atherosclerotic plaque. More oxidized and glycated LDL can increase the incidence of cerebrovascular diseases, such as stenosis, thrombosis, embolism, hemorrhage, and CVD. OxLDL is a potent inflammatory trigger of atherosclerosis and vascular complications [3,4]. Native HDL can prevent LDL oxidation by removing oxidized lipid species via the neutralization of free radicals and reactive oxygen species (ROS). An impairment of the HDL quantity and quality is associated with the incidence of CVDs, cerebral diseases [5], and kidney diseases [6]. Therefore, increasing the HDL quantity and quality could be an appropriate tool to suppress many heart and brain disorders. Regarding acute viral infections, such as coronavirus disease 2019 (COVID-19), a lower HDL quantity is associated with a high sensitivity of severe acute respiratory syndrome coronavirus 2 (SARS-CoV-2) [7] and a high risk of death [8]. HDL functionality, particularly the paraoxonase (PON-1) activity, might be important for the suppression of a SARS-CoV-2 infection [9]. A previous study reported that native HDL with antioxidant and anti-atherosclerotic activities displayed a potent antiviral activity to suppress the replication of SARS-CoV-2, while glycated HDL lost their antiviral activity [10]. Accumulated studies reported many aspects of high-density lipoproteins (HDL) in human growth, human disease, and the aging process under different environmental changes, such as dietary patterns and an exposure to pathogens, smoking, and pollutants [1,2].

## 2. Change in HDL-C Quantity during One’s Lifetime

HDL-C, usually expressed as mg/dL, can be measured directly from serum by enzymatic determination using the cholesteryl esterase and cholesterol oxidase method after precipitating apo-B containing lipoproteins. Unlikely total cholesterol (TC) and LDL-C, a low HDL-C level is one component of dyslipidemia and metabolic syndrome. Low HDL-C in men and women is defined as <40 and <50 mg/dL, respectively, according to the guidelines of the National Cholesterol Education Program Adult Treatment Panel III [11]. HDL-C can be applied to the expression of the lipid profile to indicate the risk of CVD and metabolic syndrome, such as the LDL-C/HDL-C ratio [12], triglyceride (TG)/HDL-C ratio [13], and HDL-C/TC ratio (%) [14,15]. Although the method of expression varies according to the disease (e.g., hypertension) and population cohort, HDL-C quantity is the critical factor affecting the risk of CVD and cholesterol-related disease. HDL-C quantity can change during one’s lifetime, from one’s teenage years to adulthood (approximately eighty years of age), particularly in the pubertal period, at a young age, middle age, menopausal age, and elderly age. HDL-C quantity can also be decreased as a result of individual lifestyle modifications, such as smoking, the consumption of trans fat, and intake of high fructose corn syrup.

### 2.1. Change in HDL-C Quantity between Gender during One’s Lifetime

A low serum HDL-C level in middle-aged individuals is a hallmark of metabolic syndrome [16] and a risk factor of Alzheimer’s disease [17], and vascular dementia [18] in old age. HDL-C quantity is not fixed during one’s lifetime, and can change depending on age and gender [14,19,20]. In the Korean adult population, men and women showed the highest HDL-C quantity in their 20s (49.9 ± 11.1 mg/dL and 58.1 ± 11.3 mg/dL), respectively, with a gradual decrease until reaching the lowest levels of HDL-C in their 70 s (45.7 ± 11.6 mg/dL and 50.2 ± 11.4 mg/dL for men and women), respectively [19]. The female group had a significantly higher HDL-C level (*p* < 0.001) than the male group from their twenties to seventies at 8.2 mg/dL and 4.5 mg/dL, respectively [19]. These results raised the question of when women had a higher serum HDL-C level than men.

On the other hand, both groups showed a similar HDL-C level in their 80s, approximately 45.9 ± 10.9 mg/dL and 46.6 ± 10.9 mg/dL for men and women (*p* = 0.656), respectively. The male group from their 20 s to 80 s showed a 4.0 mg/dL difference in HDL-C, while the female group showed an 11.5 mg/dL difference in HDL-C, indicating that the female group experienced a more severe decrease during their lifetime from their 20 s to 80 s [19]. The same tendency was found in studies on the American population, such as the Rancho Bernardo Study 1984–1994 [20]. This study showed that the HDL-C levels decreased in older men and women with an increase in age. The HDL-C levels decreased gradually and dramatically in individuals older than 50 years, particularly in the female group.

The dramatic decrease in HDL-C after middle age is linked to the sharp increase in dementia in people in their 80 s. The female group showed a three-fold higher level than the male group [19]. Interestingly, both genders, men and women, showed a direct association between the income grade with the HDL-C level; the lower-income group showed a higher prevalence of low HDL-C levels [19,21].

### 2.2. Pubertal Change in HDL-C in Men

Why and when men have a lower HDL-C level than women in adulthood has been a mystery for a long time. A study on teenage subjects showed that the lowering of HDL-C in men is related to the remarkable decrease in serum HDL-C during the pubertal age, 14–15 years old [22]. The male group showed a sharper decrease in the HDL-C level from 10–11 years old (54.6 ± 11.1 mg/dL, *p* < 0.001) to 14–15 years of age (48.4 ± 9.0 mg/dL), while the female group showed a similar increase in the HDL-C level from 10–11 (52.8 ± 9.6 mg/dL) to 14–15 years of age (53.1 ± 9.7 mg/dL, *p* = 0.001). At 18–19 years of age, the HDL-C level in the male group decreased to 50.3 ± 9.3 mg/dL, whereas the female group showed an increase to 55.6 ± 10.5 mg/dL. HDL can cross the blood–testis barrier (BTB) to supply cholesterol for spermatogenesis in the male group, while LDL cannot. These results suggest that the dramatic gap between the HDL-C between males and females was initiated from the pubertal age, especially in the age range of 14–15 years. The male group has a remarkable demand for cholesterol for spermatogenesis in the testis, which explains the sharp decrease in HDL-C levels during the pubertal period for boys.

A higher HDL-C level is inversely associated with a lower incidence of cardiovascular disease [23]. Generally, women have a higher HDL-C than men between 20 and 50 years of age, particularly before experiencing the menopause. Other reports showed that women had a 5 mg/dL higher HDL-C level than men from the unadjusted mean difference in HDL-C [24], although it was unclear when and why men have a lower HDL-C level than women in adulthood.

It is essential to compare the HDL-C level in teenagers (10–19 years old) between boys and girls to understand when and why women have a higher HDL-C level and a longer life span than men in adulthood and later life [25]. The HDL-C level rapidly decreases during the pubertal period (14 and 15 years old) in the male group. The lowest HDL-C level at 15 years of age was not restored at 19 years of age in the male group; it was almost fixed and remained at a lower level for the life expectancy of men compared to women. The decrease in HDL-C during the pubertal age occurred only in the male group, but the reason for this is unclear. One possible explanation is that the cholesterol from HDL is required to produce the male hormone and spermatogenesis in boys. Cholesterol is essential for spermatogenesis and steroidogenesis in the male reproduction system. Sertoli cells, which promote sperm production, block the passage of LDL at the blood–testis barrier (BTB), but permit the entry of HDL to the seminiferous tubules [26]. Cholesterol is an essential nutrient for spermatogenesis because it fuels spermatogenesis in Sertoli cells. The lower HDL-C during the teenage period in males might be associated with the lower life expectancy of men in adulthood and remaining later life. Furthermore, the lower HDL-C in men might be correlated with the higher BP in men during adulthood [14].

### 2.3. Menopausal Change of HDL-C in Women

The menopause is a critical event in a woman’s life in regard to the physical and emotional changes, accompanied by a decline in ovarian activity. Despite the conflicting data [27], many studies suggested that a decrease in HDL-C is associated with the menopause [28]. In a study on the Korean population, women showed a stable HDL-C level from 58 ± 11 mg/dL to 56 ± 12 mg/dL in their 20 s and 40 s, respectively, only a 2 mg/dL difference. On the other hand, women showed a sharp decrease in HDL-C from 55 ± 12 mg/dL to 46 ± 10 mg/dL in their 50 s to 80 s, respectively, a 9 mg/dL difference. A 12 mg/dL difference in the HDL-C was detected in the female group in their 20 s to 80 s [14,19]. On the other hand, men showed a more stable and smaller decrease in HDL-C with the increase in age; the male group showed only a 4 mg/dL difference from their 20 s to 80 s [19].

Generally, post-menopausal women have an elevated cardiovascular risk via increased hyperlipidemia [29], with a two-to-three-times higher CVD death rate than the premenopausal women of the same age [30]. A meta-analysis showed that the menopause effect was associated with around a 10–20% increase in TC, LDL-C, and triglycerides (TGs), while only HDL-C decreased by around 10% [27]. Interestingly, several studies suggested that post-menopausal women have an increased level of apoA-I (a 13% increase) [31], despite the decrease in HDL-C [32,33]. These discrepancies suggest that the HDL particle-formation ability of apoA-I in post-menopausal women might be impaired. Lipid-free apoA-I could be increased in post-menopausal women, despite the similar kinetic parameters of HDL and apoA-I in the two groups [34]. Post-menopausal women showed a greater decrease in large-sized HDL (~22%), with an increase in lipoproteins (Lps) containing apoA-II (LpA-II), than premenopausal women, while the LDL size was similar in the two groups [34]. These results suggest that the menopause is a critical event in a woman’s life that changes the HDL quantity [27] and quality [35], similar to puberty playing a critical role in changing HDL-C in a man’s life.

### 2.4. Exercise and Change in HDL

A sedentary lifestyle is associated with low HDL-C for all ages and genders [36,37]. A meta-analysis with 19 randomized controlled trials showed that exercise could improve the cardiometabolic risk factors, lowering the TC, LDL-C, and TGs in obese adolescents with sedentary behavior [37]. Interestingly, the increase in HDL-C as a result of exercise was prominent in studies on obese children [38]. Although short-term exercise (less than six months) could not sharply decrease the total cholesterol, HDL-C was increased as a result of exercise in healthy children [39]. Another report showed that the performance of resistance exercises for 14 weeks was not associated with improved HDL-C, even though the TC and LDL-C decreased in premenopausal women [40]. There are conflicting data concerning the effect of exercise on improving HDL-C quantity, depending on the type of sport [41]; aerobic exercise effectively improves HDL-C, but resistance exercise does not.

Aerobic exercise performed for 12–24 weeks can increase HDL-C more efficiently, by around 3.8–15.4 mg/dL from the initial level [42]. A recent meta-analysis also showed that aerobic exercise was the best option to increase HDL-C [43]. In the same context, in Olympic athletes, aerobic exercise (runners and wrestlers) showed a 1.3-fold higher HDL-C level than anaerobic exercise performed by elite athletes [44]. They presented a larger HDL particle size, higher apoA-I content, and higher antioxidant enzyme (paraoxonase) activity. These results suggest that performing repetitive aerobic exercise can increase HDL-C quantity and the HDL quality.

### 2.5. Nutritional Supplementation and the Change in HDL

#### 2.5.1. Omega-3 Consumption

The elevation of HDL-C quantity by the consumption of omega-3 is still controversial, depending on a combination of eicosapentaenoic acid (EPA) and docosahexaenoic acid (DHA) and the consumption period. The consumption of 2.8 g/d EPA and 1.7 g/d DHA for 6 weeks resulted in a selective increase in cholesterol in the subfraction of larger HDL (HDL_2_) by up to 74%, with a concomitant 19% decrease in HDL-C [45]. However, the reason why only cholesterol in HDL_2_ was increased, despite the lack of change in the TC, HDL-C, and TGs, is still unclear. A meta-analysis of the data from 33 randomized controlled trials reveals that dietary EPA supplementation on metabolic syndrome did not increase HDL-C (weighed mean difference, WMD = 0.02 nmol/L), while DHA supplementation increased HDL-C (WMD = 0.07 nmol/L) [46]. In contrast to the inconsistent results obtained from the combined supplementation, the supplementation of omega-3 (marine n-3) alone (1 g/d, n = 12,933) for 5.3 years did not reduce the risk of major cardiovascular events than the placebo without an increase in HDL-C, compared to the vitamin D3 (2000 IU/d) group (n = 12,938) as a placebo [47]. In the long-term and larger participant study, the high-dose consumption of icosapent ethyl (4 g/d) for 4.9 years decreased the risk of ischemic events and increased HDL-C [48].

The consumption of high-dose omega-3 fatty acids (4 g/d, n = 6539) for 12 months resulted in no significant changes in HDL-C (from 36 mg/dL at the baseline to 37 mg/dL at the follow-up in statin-treated participants with a high cardiovascular risk [49]). In comparison to the corn oil group (n = 6539) as a placebo, the geometric mean ratio between the groups was 1.01 (*p* = 0.002), indicating that HDL-C increased very slightly. Interestingly, the serum apo-B content was not changed between the groups, suggesting that the quality of HDL and LDL might not be changed. More recently, the consumption of omega-3 krill oil (4 g/d) for 26 weeks resulted in a decrease in the TG, but no significant improvement in HDL-C [50]. Therefore, despite the conflicting data, it can be concluded that the consumption of omega-3 did not increase the serum HDL-C in both long-term and short-term consumptions.

#### 2.5.2. Consumption of Policosanol from Sugarcane Wax

Policosanol is a mixture of aliphatic alcohols, ranging from 24–34 carbon atoms, such as octacosanol, triacontanol, dotriacontanol, hexacosanol, and tetratriacontanol as the major components, which were purified from sugarcane (*Saccharum officinarum* L.) wax or various plants, rice bran, and beeswax. A meta-analysis of 22 studies reported that policosanol could lower the lipid content and is a safe drug used to elevate HDL-C levels [51].

Cuban policosanol in reconstituted HDL (rHDL) showed potent cholesteryl ester transfer protein (CETP) inhibition activity from HDL to LDL, as shown in previous reports [52,53], even though the policosanol in ethanol did not show adequate activity. Policosanol in rHDL possesses a much higher CETP inhibitory ability, around 67% inhibition, than rHDL alone, which showed a 10% inhibition [52,53]. In the molecular structure, policosanol contains long-chain hydrophobic moieties, which can interact with the hydrophobic active site of CETP. The aliphatic chains can interfere with binding between CETP and HDL. The aliphatic chains bind to a putative CE-binding site in the C terminus of CETP to form a ternary complex, as previously suggested [54]. Other in vitro and animal studies reported that policosanol inhibits cholesterol synthesis by modulating 3-hydroxy-3-methylglutaryl-coenzyme A (HMG-CoA) reductase, possibly by activating 5′ adenosine monophosphate-activated protein kinase (AMP-kinase) [55]. Other studies determined that the action of policosanol involves various pathways, such as the activation of AMP kinase, stimulation of the Akt signaling pathway to prevent lipopolysaccharide (LPS)-induced apoptosis [56], down-regulation of HMG-CoA reductase, and cholesteryl ester transfer protein inhibition [57,58,59]. In addition to the increase in HDL quantity and improvement of dyslipidemia, the consumption of Cuban policosanol also enhanced the HDL functionality [59] and the blood-pressure-lowering effects [60,61].

## 3. HDL Quality

HDL quality is a more diverse and complex concept than HDL quantity. It is defined in the following manner: “HDL morphology, the composition of lipid and protein, and functionality include particle shape and number, glycation and oxidation extent, antioxidant activity, and cholesterol binding and efflux ability” [1,2].

### 3.1. HDL Particle Shape, Size, and Antioxidant Ability

HDL quality is a general feature of the particle morphology consisting of the particle shape, size, and numbers based on an observation of transmission electron microscopy (TEM). A premature HDL in the early stage of biogenesis is a disc shape, and a mature HDL is a spherical shape. The features of a desirable HDL are its distinct particle shape, larger size, and greater numbers. The beneficial HDL quality is related to the healthier LDL quality and less oxidation and glycation, which are directly associated with CVD risk. The worst HDL, glycated HDL by a fructose treatment, showed the smallest particle size (~9.6 ± 1.5 nm) with aggregated and unclear particle morphologies. The glycation caused a change in the HDL particle shape into a more amorphous and fibrous shape, such as sticky amyloid fiber.

Recently, a cross-sectional study in the population-based LifeLines DEEP cohort reported that large HDL particles were negatively associated with the leukocyte counts and were linked to a decreased cardiovascular risk [62]. Because the expansion of white blood cells has been linked to increased CVD [63] and rheumatoid arthritis (RA) [64], good HDL quality is important for suppressing the incidence of CVD and RA. RA patients presented bad quality HDL; a decreased particle size and number of HDL with reduced paraoxonase activity [65]. Paraoxonase is a critical antioxidant enzyme in HDL with lactonase and ester hydrolase, which degrades the lipid peroxides in LDL and inhibits viral infections [10].

In regard to the anti-infective ability of HDL, an improvement in the antioxidant capacity of HDL has been associated with the eradication of the hepatitis C virus (HCV) [66]. Low blood levels of paraoxonase-1 (PON-1) are associated with more severe symptoms in patients with viral hepatitis [67] and poor survival in patients with severe sepsis [68]. A study on Chinese patients with COVID-19 showed that a decrease in HDL-C was positively correlated with the severity of COVID-19 [69]. A meta-analysis of twenty-four studies revealed a correlation between low HDL-C and the severity of COVID-19; low serum HDL-C levels were a poor prognostic factor for the severity of COVID-19 [70]. Native HDL with a sufficient paraoxonase activity displayed potent antiviral activity to suppress the replication of SARS-CoV-2, while glycated HDL lost their antiviral activity [10]. Glycated HDL were a smaller particle size and presented a loss of PON activity to display the atherogenic properties. The pro-inflammatory effects of modified HDL with a loss of PON activity were mediated by cellular signaling between the blocking of scavenger receptor-B-I (SR-B-I) and the binding of glycated HDL [71].

### 3.2. ApoA-I Content in HDL

Apolipoprotein A-I (apoA-I), the major protein of HDL, exerts antioxidant and anti-inflammatory activities in lipid-free and lipid-bound states, along with cholesterol efflux activity [72]. ApoA-I is a major protein component, around 70%, in HDL and exerts antioxidant activity [73]. The full length of the polypeptide (243 amino acids) and higher apoA-I content is linked to the better quality of HDL, because the particle size, number, and shape of HDL are strongly dependent on the apoA-I structure. Because the HDL particle number and size are good predictors of the CVD risk, a higher apoA-I content in HDL is negatively associated with the CVD risk [74]. Recently, instead of LDL-C/HDL-C, the apo-B/apoA-I ratio has emerged as an important marker to predict long-term major cardiovascular events [75]. The reduction in apoA-I is a remarkable risk factor for cardiovascular events, because apoA-I regulates the HDL quality.

### 3.3. ApoA-II Content in HDL

ApoA-II is the second most important constituent of HDL apolipoproteins and exists as a homodimer with 2 polypeptide chains, each having 77 amino acids in length [76]. An increased serum apoA-II level might be associated with the development of atherosclerosis, with increased indices of atherogenic lipoproteins and combined hyperlipidemia [77]. Furthermore, human apoA-II enrichment in an HDL displaces paraoxonase from the HDL and disrupts or inhibits its antioxidant properties [78]. Therefore, apoA-II is considered an atherogenic factor that hinders the beneficial functions of HDL, including the antioxidant ability.

Commercial HDL contain two bands, around 25 kDa (major) and 10 kDa (minor) for apoA-I and apoA-II, respectively [79]. As claimed by the manufacturer, according to the electrophoretic profile (12% Tris-Glycine gel), the minor band is apoA-II. ApoA-II is a homo-dimeric protein around 17.4 kDa that can be reduced to a monomeric protein (around 8.7 kDa). In commercial HDL, apoA-I and apoA-II comprise around 70% and 20% of the total HDL protein content, respectively. The HDL containing apoA-I and apoA-II facilitated viral entry via SR-B-I with a loss of antiviral activity [79]. On the other hand, purified native HDL via ultracentrifugation using the conventional method showed potent antiviral activity against SARS-CoV-2.

ApoA-II might modulate the interaction of HDL and scavenger receptor-B-I (SR-B-I). The apoA-II level in HDL was inversely associated with HDL binding and selective cholesteryl ester (CE) uptake by SR-B-I [80]. Although an association between apoA-II and the risk of coronary artery disease (CAD) has been controversial, the apoA-I level is inversely associated with the risk of CAD and CVD. Human apoA-II may also be involved in the displacement of apoA-I in a reconstituted HDL, which impairs HDL functionality [81]. Moreover, apoA-II enrichment in an HDL displaces the paraoxonase from the HDL and disrupts or inhibits its antioxidant properties [78,81].

Interestingly, heavy alcohol drinkers (54.9 ± 23.3 g of alcohol/day) showed a 15% higher HDL-C level than non-drinkers, even though the apoA-II level was also higher in the heavy-drinker group (33.7 ± 6.7 mg/dL) than that of the non-drinkers (29.1 ± 4.4 mg/dL) [82]. This result shows that heavy-drinker HDL quality is undesirable via a greater enrichment with LpA-I:A-II, which is more atherogenic, but the HDL-C level is increased. Lipoproteins (Lps) containing apoA-I alone (LpA-I) and Lps containing both apoA-I and apoA-II (LpA-I:A-II) are structurally and metabolically different [83]. LpA-I:A-II is more pro-atherogenic than LpA-I [84]. Overall, an increase in the apoA-II content impaired the HDL quality and functionality, regardless of the elevation of the HDL-C level.

### 3.4. ApoC-III Content in HDL

ApoC-III is a primary component of the triglyceride-rich lipoproteins, such as chylomicron, and very low-density lipoproteins (VLDL), and its concentration is highly and positively correlated with the serum TG concentration [85]. As the apoC-III content is increased in HDL, HDL might be impaired functionally and structurally to have more atherogenic properties. Patients with CAD also showed a significant elevation of apoC-III in HDL [86], despite the similar plasma apoC-III level between the CAD and control. Interestingly, the elderly group showed a greater increase in apoC-III in HDL with a smaller particle size than the young control [87]. The elevated apoC-III content was linked to the senescence-related truncation of apoA-I and an increased level of glycation in the HDL in the elderly group. Overall, the increase in apoC-III and the smaller molecular weight of apoA-I in HDL can be associated with the production of dysfunctional HDL in vitro [88] and smokers’ HDL [89].

Regarding the energy metabolism, high concentrations of apoC-III blocked oxidative phosphorylation and oxygen consumption and inhibited the state III respiration in the mitochondria [90]. ApoC-III induced sterile and systematic inflammation via the activation and release of interleukin (IL)-1β, IL-6, and tumor necrosis factor (TNF)-α [91]. Moreover, apoC-III also caused organ damage by alternative inflammasome activation and was associated with increased mortality, particularly in patients with an acute myocardial infarction and chronic kidney disease [91].

Furthermore, the rapid elevation of apoC-III in HDL is associated with an acute viral infection, such as hemorrhagic fever renal syndrome (HFRS) [92]. During a Hantaan viral infection, the serum HDL-C level is remarkably reduced in the oliguric phase with severe acute inflammation [93]. Although the precise mechanism is unknown, the elevated apoC-III might be associated with the infection and replication of Hantavirus.

### 3.5. α-Synuclein, β-Amyloid, and Serum Amyloid A in HDL

The aggregation of α-synuclein (α-syn) is a major culprit of Parkinson’s disease (PD), but the role of α-syn in the brain lipoprotein metabolism is critical in the pathogenesis of PD. The metabolism of cholesterol and lipoprotein in the brain plays an important role in the action of α-syn, a major component of Lewy bodies, in the pathogenesis of PD [94]. Low serum apoA-I and HDL-C levels are correlated with an earlier onset of PD [95]. Serum HDL can interact with α-syn to inhibit the aggregation of α-syn in the brain [96]. Therefore, a putative interaction between α-syn and apoA-I might regulate cholesterol homeostasis. Purified α-syn and apoA-I can interact to maintain a solubilized HDL complex form and impair the structural and functional features of HDL [97].

Beta-amyloid (Aβ) is a causative protein in the pathogenesis of Alzheimer’s disease (AD). Regarding the brain lipid metabolism of AD pathophysiology, reduced serum apoA-I and elevated serum apoC-III have been detected in AD [98]. Because apoA-I and discoidal HDL can cross the blood–brain barrier (BBB), the metabolism of apoA-I and HDL is essential for the clearance of Aβ. ApoA-I and discoidal HDL in the blood can be transported into the brain tissue via clathrin-independent and cholesterol-mediated endocytosis pathways [99]. The beneficial functions of apoA-I and HDL were severely impaired by the addition of Aβ via its detrimental effect on the secondary structure [100]. The impairment of HDL functionality occurred synergistically by the co-addition of fructose and Aβ. Mild cognitive impairment (MCI), AD, and dementia in older age are well correlated with low serum HDL-C levels, particularly in middle age [17,101].

Human serum amyloid A (SAA) is an acute-phase reactant that is related to CVD, whose levels increase because of an acute infection, inflammation, and tissue damage in the blood [102]. Human SAA, a family of apolipoproteins associated with high-density lipoproteins (HDL) in the plasma, is secreted during the acute phase of inflammation and functions as the means of transport of cholesterol to the liver for secretion into the bile [103]. Human SAA, consisting of 103–104 amino acids, is the precursor protein in inflammation-associated amyloid A [104]. The major functions of human SAA are associated with amyloidogenesis, the remodeling of HDL, tumor pathogenesis, anti-bacterial functions, and regulation of inflammation and immunity [105]. In sepsis, patients in the critical phase showed that the HDL-C level decreased to approximately 14 mg/dL, while SAA sharply elevated to around 2827 μg/mL, in contrast to the control group that showed a normal level of HDL-C (~39 mg/dL) and 269 μg/mL of SAA [106]. SAA is expressed almost exclusively in the liver and is triggered by pro-inflammatory cytokines. Newly synthesized SAA displaces apoA-I and becomes an apolipoprotein of HDL [104]. SAA-enriched HDL are prone to becoming dysfunctional HDL and releasing lipid-free apoA-I [107], which helps to generate poor quality HDL. The low HDL-C levels are related to the elevated levels of interleukin-6 and other cytokines, namely the “cytokine storm” and increased hepatic production of SAA during inflammation.

### 3.6. Lipid Compositions in HDL

In addition to the protein component, lipids, such as cholesterol and the TG, also influence HDL quality. A higher cholesterol content in HDL induces a larger particle size, whereas a higher TG content in HDL induces a smaller particle. TG-enriched HDL also impaired the functionality to promote abnormal cholesterol transport and efflux. Smokers showed a smaller particle size with a higher TG content than non-smokers at the same age, even though they were young and light smokers, smoking less than five cigarettes per day [89]. The HDL of smokers, which were more oxidized and glycated, exacerbated the cellular senescence and atherogenic process [89]. In addition to the lower cholesterol content in HDL, a higher TG content in HDL is associated with a higher incidence of metabolic syndrome via poor HDL quality. The serum TG and the TG/HDL-C ratio were associated with hypertension in non-hypertensive Middle Eastern women [108].

### 3.7. The Best HDL versus the Worst HDL

The healthiest HDL are enriched with cholesterol and apoA-I with a distinct round shape and a larger particle size, as illustrated in Figure 1. However, a healthy HDL can be impaired by many stressors, such as aging, smoking, infection, pollutants, and poor food habits (the consumption of trans fats and fructose corn syrup). The worst HDL showed the displacement of apoA-I and a decrease in cholesterol and enrichment with TG, SAA, and apoC-III. The particle morphology changed to a smaller size and ambiguous shape by exposure to glycation stress, smoking, acute infection, and oxidative stress, as shown in Figure 2. A transmitted electron microscopy (TEM) image of glycated HDL by a fructose treatment showed a smaller particle size and ambiguous morphology without a distinct particle shape [10]. The glycated HDL also showed a loss of PON activity, with a decrease in the particle size and ambiguous morphology [10].

## 4. HDL Functionality

### 4.1. Overview

The functionality of HDL, mainly cholesterol efflux ability and antioxidant activity, is essential for suppressing cardiovascular disease, stroke, and dementia. More recently, coronary heart disease has been reported to be a potent risk factor for dementia [109,110]. Many cardiovascular risk factors, such as hypertension, type 2 diabetes mellitus, and dyslipidemia, are also significant risk factors of AD and vascular dementia [111]. High dysfunctional HDL and low serum apoA-I levels are hallmarks of diabetes, inflammation, and CVD [112]. Higher serum apoA-I levels and enhanced HDL functionality are essential for suppressing CVD [113].

Regarding carbohydrate toxicity to impair HDL functionality, diabetes mellitus and AD are associated with the pathogenesis of advanced glycated end (AGE) products [114]. Glycated apolipoproteins are associated with a higher incidence of diabetes and AD [115], but the mechanism is unclear. HDL functionality is strongly dependent on the composition of apolipoproteins and their extent of glycation [116]. HDL functionality, including cholesterol efflux and antioxidant and anti-glycation activity, can be enhanced by aerobic exercise [117] and healthy nutrition [56,57,58,59,60].

### 4.2. Cholesterol Efflux and the Treatment of Dyslipidemia via Reverse Cholesterol Transport (RCT)

The basic functionality of HDL is a receiving ability for excess cholesterol from peripheral cells to form discoidal HDL, which are an early type of HDL, so-called cholesterol efflux activity in reverse cholesterol transport. Nascent HDL can take up cholesterol from macrophages in the artery wall via the ATP-binding cassette A-1 receptor (ABCA-1) [118]. Lipid-free apoA-I can bind to cholesterol in blood cholesterol pumps (ABCA1 and ABCG1) in peripheral cells, and macrophages contribute to the maturation of HDL via the efflux of cholesterol.

Lipid-free apoA-I binds to cholesterol and forms discoidal HDL in the early stage of cholesterol efflux, as shown in Figure 2. The further binding of free cholesterol and esterification to the cholesteryl ester via lecithin:cholesterol acyltransferase (LCAT) accumulates a core of spherical HDL. Once associated with phospholipid (PL), free cholesterol (FC) and apoA-I form disc-shaped HDL particles for growth and maturation. The higher cholesterol efflux and esterification of cholesterol to cholesteryl ester (CE) can produce spherical HDL from discoidal HDL. During the maturation process, with the accumulation of cholesteryl ester, the HDL_3_ (smaller particle) grows to HDL_2_ (larger particles), as illustrated in the central part of Figure 2. In the blood stream, there are two major subfractions of HDL, HDL_2_ and HDL_3_. HDL_3_ is an early type of HDL and enriched with protein with a smaller particle size. HDL_2_ is a matured type of HDL and enriched with cholesterol with a larger particle size.

On the delivery end of the pathway, apoA-I is important because it provides HDL cholesterol for the steroid hormone and bile acid synthesis in the adrenals and liver via interactions with the scavenger receptor type B class I (SR-B-I), as shown at the top of Figure 2 [119]. Alternatively, HDL cholesterol can be transferred to apo-B-containing proteins (LDL, IDL, and VLDL), via the action of the cholesteryl ester transfer protein (CETP) in the bloodstream. The higher CETP activity resulted in the recycling of cholesterol and an increasing attenuation in blood circulation, potentially contributing to build-up on the artery wall [120]. As depicted in Figure 2, a higher CETP activity induced the elevation of LDL-C and production of oxLDL via a greater attenuation of cholesterol in LDL. The oxLDL are prone to accumulate in atherosclerotic plaque (Figure 2). Thus, CETP is recognized as an atherogenic factor that can facilitate oxLDL production and be taken into macrophages to build atherosclerotic plaque.

The serum HDL can interact with β-amyloid (Aβ) in the brain [121] because apoA-I can cross the BBB, as shown in Figure 2. Human apoA-I exhibited a binding ability with Aβ to prevent Aβ-induced neurotoxicity [122]. Because apo-B cannot cross the BBB, there is no LDL particle in the brain side, indicating that cholesterol in the blood side cannot transport into the brain side.

### 4.3. Treatment of Alzheimer’s Disease via Amyloid Removal

HDL have emerging roles in neurodegenerative disorders because cholesterol transport in the brain is strongly dependent on HDL metabolism, not LDL. The higher plasma levels of HDL and apoA-I are directly correlated with a lower risk of AD and dementia [123]. Although the accumulation of Aβ is a major culprit of AD, reduced serum apoA-I and elevated serum apoC-III have been detected in AD patients [124]. Although the specific mechanism has not been fully elucidated, as illustrated in Figure 3, apoA-I has a protective ability in the AD pathology, including ameliorating Aβ deposition and memory reduction [99,125]. There are two types of HDL, apoE-HDL (HDL containing only apolipoprotein E) and apoA-I-HDL, in the brain side because apoA-I can cross the BBB, and brain glial cells can synthesize apolipoprotein E (apoE) for their brain cholesterol transport (Figure 3). ApoA-I-HDL can bind with Aβ and excess cholesterol from brain cells and transfer back to the blood side for excretion, as shown in Figure 3. The in vitro binding of Aβ with HDL impaired the structure and functionality of the HDL to result in the aggregation of apoA-I and the loss of anti-atherogenic activity [100]. In addition to the inhibition of the oligomerization of Aβ, HDL and apoA-I can inhibit the aggregation of Aβ, in the process of amyloid plaque formation, as shown in the bottom section of Figure 3.

The upregulation of the HDL-C level is helpful for preventing vascular dementia, which is the second most common cause of dementia [111,126]. Elevating HDL-C and enhancing HDL functionality would help reduce the oxidized species (4-hydroxynonenal and inducible nitric oxide synthase), inflammatory cytokines (IL-6 and TNF-α), and amyloid plaque size. Many reports explain the relationship between the higher serum HDL-C in middle-aged individuals and the lower incidence of dementia in late age [127].

### 4.4. Treatment of Hypertension via the Regression of Plaque and Reduction in Aortic Stiffness

Hypertension is a major risk factor for developing CVD, which is frequently accompanied by other risk factors, such as dyslipidemia and diabetes [128]. The serum HDL-C is inversely correlated with the incidence of aging-related diseases, such as CVD, diabetes, and AD [129,130]. The HDL-C/TC ratio (%), rather than HDL-C (mg/dL), is more important for predicting the risk of incident hypertension [131]. The quality and functionality of HDL is also an important factor in suppressing atherosclerotic progression. The impaired functionality of HDL, such as cholesterol efflux, is closely associated with atherosclerotic CVD [132].

Hypertension is closely linked to the incidence of metabolic syndrome [133], which involves abdominal obesity, high serum TG level, low HDL-C level, and insulin resistance. A higher quantity of VLDL increases aldosterone production in the mitochondria by binding SR-B-I and stimulating several signaling pathways in acute aldosterone secretion and sustained aldosterone production [134]. Modified LDL, such as oxidized LDL and glycated LDL, also stimulate the release of aldosterone via Jak-2 activation for adrenocortical steroidogenesis [135]. These results suggest that the elevated LDL-C, oxLDL, and dysfunctional HDL cause higher BP via greater aortic stiffness and aldosterone secretion. As shown in Figure 4, although there are many mechanistic interpretations to explain the pathogenesis of hypertension, one of the basic mechanisms is vasculopathy, which is a narrowed lumen by building up atherosclerotic plaque.

In the higher blood pressure state, the left side of Figure 4, LDL are easily oxidized and glycated to oxLDL and small dense LDL (sd-LDL). HDL are enriched with TG, SAA, and apoC-III to be dysfunctional HDL, which cannot prevent the oxidation and glycation of LDL. The oxLDL phagocytosis is more accelerated and increases the plaque size because the dysfunctional HDL act as bystanders or cause even more exacerbation. The oxLDL and sdLDL accumulate to build more atherosclerotic plaque and create a narrower lumen and stiffened aorta, which allow for the elevation of BP.

The HDL functionality is improved by nutritional supplementation. Exercise produces good HDL that are enriched with cholesterol and apoA-I to inhibit the oxLDL phagocytosis and CETP activity, as shown in the middle section of Figure 4. The prevention of the atherogenic process and regression by enhanced HDL functionality help to alleviate high BP in blood vessels. Regression is accomplished by healthier HDL to exert cholesterol efflux and inhibit LDL oxidation, resulting in more cholesterol excretion from atherosclerotic plaque. The HDL grows to a larger HDL, enriched with cholesterol and apoA-I, and easily binds to scavenger receptors for excretion via the liver, as shown in the right part of Figure 4. Finally, the BP can be lowered by widening the lumen and relieving aortic stiffness.

### 4.5. Application of HDL

The quantities, qualities, and functions of HDL are an emerging area of recent research to develop new diagnoses and therapeutics. The physiological defense mechanisms of HDL against oxidizing and inflammatory attacks might be determined. The composition of apolipoproteins and lipids may influence the quality and functions of HDL. Improved HDL function and quality allows for the generation of more favorable outcomes, with regard to the augmentation of HDL-cholesterol levels, thereby allowing enhancements of the innate therapeutic functions associated with HDL [1].

Native HDL have many beneficial functions, such as anti-atherosclerosis, anti-hypertension, and anti-infection activity, as well as anti-bacterial and antiviral, particularly anti-COVID-19, as shown in Figure 5. Determining the HDL quantity and quality can be applied to develop new diagnostic biomarkers, such as apoA-I content, extent of glycation, particle size, and number in HDL. The HDL functionality can be applied to develop a new protein drug for regression and rejuvenation and a delivery vehicle for gene therapy and non-soluble drugs [2].

## 5. Conclusions

The quantity of HDL-C is inversely correlated with the incidence of CVD. Although an extremely high level of HDL-C is associated with an increased risk of death, a mildly higher HDL-C level is also associated with the lowest risk of death. These discrepancies suggest that the individual HDL-C quantity cannot explain disease risk and expectation of longevity. In addition to the HDL-quantity, the HDL-quality and its functionality are also essential to lower the risk of CVD, hypertension, and dementia. Patients with diabetes or CVD showed dysfunctional HDL, which are more oxidized and glycated, with lower cholesterol and apoA-I content. Dysfunctional HDL have a poor HDL quality, lower antioxidant ability, ambiguous particle morphology, and smaller size. On the other hand, the CETP activity and expression content are relatively higher in dysfunctional HDL, even though the PON activity disappeared.

The desirable features of HDL are those that are enriched with cholesterol and apoA-I with a smaller amount of CETP and larger particle size. Its functionalities extend to anti-glycation, anti-inflammatory, and antioxidant activities to attenuate aging stress. A desirable HDL status can be maintained by exercise and avoiding stress to protect it from oxidation, glycation, and environmental stresses, because the features of the HDL structure and functionalities can be changed depending on the health and disease state.

In this review, various aspects of HDL were discussed, including their structure and function correlations, their protein components and roles, impairment of HDL, and strategies for improving HDL quantity and quality. HDL are a valuable target in treatments for atherosclerotic hypertension and CVD, including CETP inhibitors, peroxisome proliferator-activated receptor (PPAR)-α agonist, PPAR-γ agonist, and HDL blood-infusion therapies. In addition to the cardioprotective effects, the antioxidant ability of HDL and apoA-I is an essential property for suppressing acute infection, inflammation, and senescence-related neurodegenerative disorders, such as Alzheimer’s disease and vascular dementia.

## Figures and Tables

**Figure 1 ijms-23-03967-f001:**
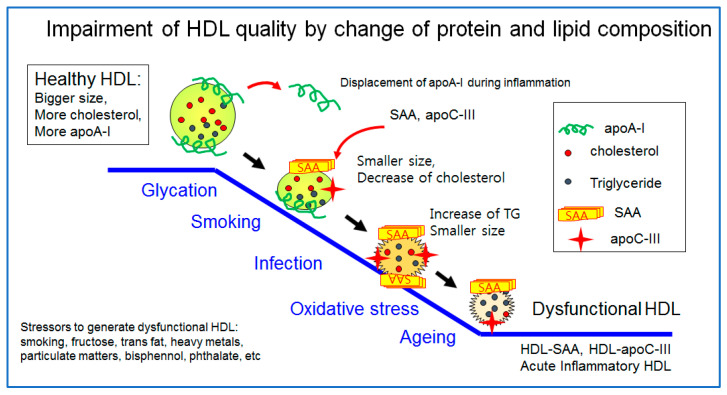
Impairment of HDL quality by exposure to glycation stress, smoking, acute infection, and oxidative stress. Healthy HDL showed a larger size with the enrichment of cholesterol and apoA-I. On the other hand, poor HDL showed a smaller particle size with an increase in the TG content, SAA, and apoC-III.

**Figure 2 ijms-23-03967-f002:**
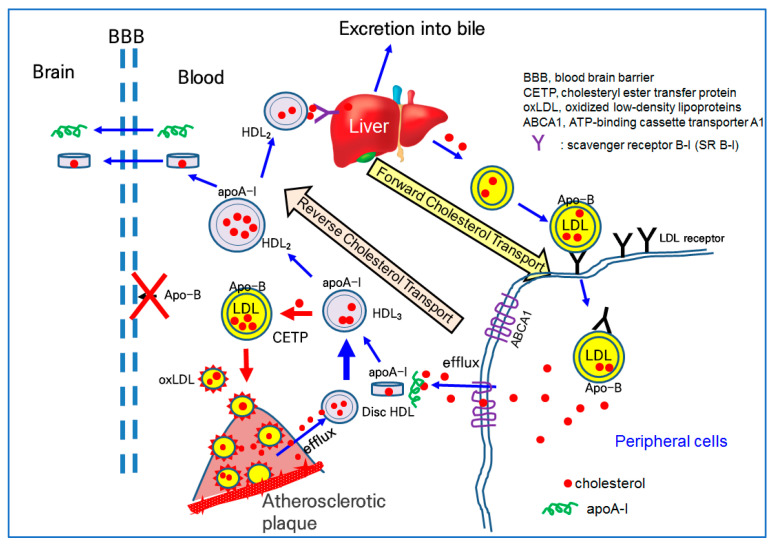
Synthesis, delivery, and excretion of cholesterol in the blood via forward cholesterol transport (FCT) and reverse cholesterol transport (RCT). LDL and HDL are the main carriers of FCT and RCT, respectively. CETP transfers CE from HDL to LDL, resulting in the decrease in the cholesterol in HDL and elevation of LDL-C.

**Figure 3 ijms-23-03967-f003:**
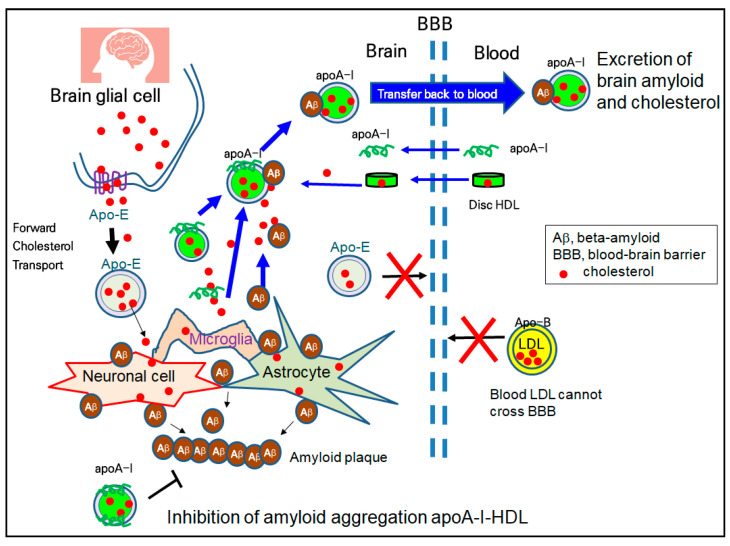
Synthesis, delivery, and excretion of cholesterol and Aβ in the brain via the apo-E-HDL metabolism (similar to forward cholesterol transport) and apoA-I-HDL metabolism (similar to reverse cholesterol transport). ApoA-I-HDL and lipid-free apoA-I can cross the BBB back and forth, while apo-E-HDL or apo-B-HDL cannot cross the BBB.

**Figure 4 ijms-23-03967-f004:**
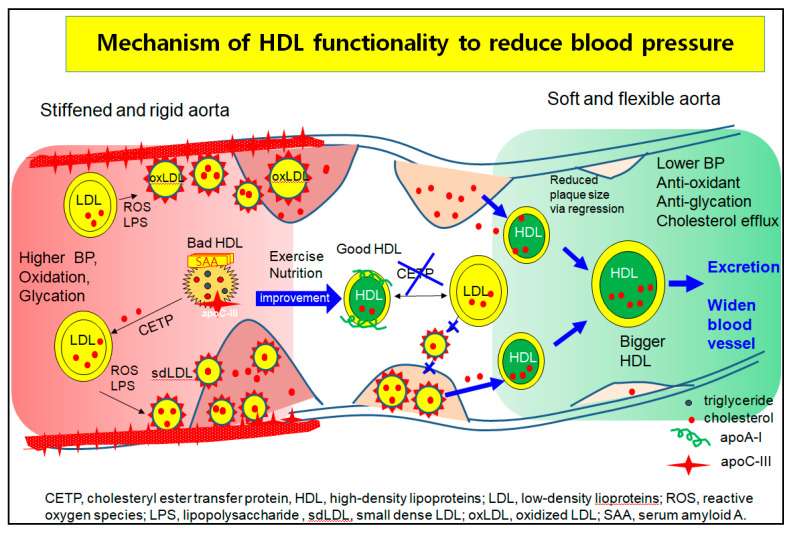
Mechanism of vasculopathy in hypertension and its treatment by enhanced HDL functionality. Oxidized LDL and dysfunctional HDL exacerbated the atherogenic process to cause hypertension. Dysfunctional HDL are enriched with TG, SAA, and apoC-III, while good HDL are enriched with apoA-I and cholesterol.

**Figure 5 ijms-23-03967-f005:**
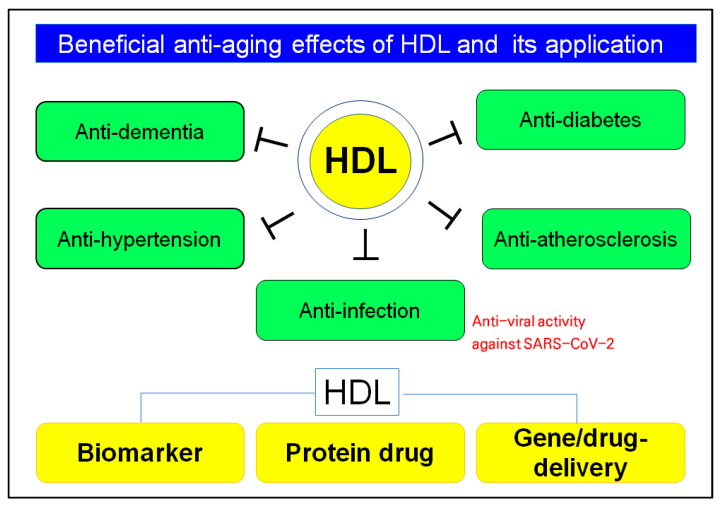
Beneficial effects of HDL quantities, qualities, and functionalities to prevent infection, inflammation, and aging-related disease. Many applications of HDL particles as diagnostic biomarkers, protein drugs, and delivery vehicles for insoluble drugs.

## Data Availability

The data used to support the findings of this study are available from the corresponding author upon reasonable request.

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
