# Peer review of "The Current Status of Research on High-Density Lipoproteins (HDL): A Paradigm Shift from HDL Quantity to HDL Quality and HDL Functionality"

_ijms, 2022, doi:10.3390/ijms23073967_

Round 1
Reviewer 1 Report
Dear Author,
I would like to congratulate you on the submitted manuscript. The analysis on HDL quantity, quality, and function is highly relevant in the actual context, and your review gives a great overview of the topic.
While reading your manuscript, I have, however, observed minor aspects that could improve its quality even further. Please consider revising the following aspects:
Line 24: “the aggregation of amyloidogenesis” should be rephrased, as aggregation leads to amyloidogenesis.
Abstract: The word “good” might be slightly overused. Therefore, please consider replacing it in instances such as “… HDL can be a good biomarker …” (Line 33. Synonym - suitable).
Line 35: “anti-inflammation abilities” you could consider replacing with “anti-inflammatory abilities”.
Line 43: “What is HDL quantity, quality, and functionality?” please rephrase in plural.
Line 50: When referring to quality, the number of particles is quantifiable, and not a morphological trait. Please review.
Line 56: “oxidized LDL (oxLDL) and small dense LDL 56 (sdLDL)” are not expressed as properties of LDL. The oxidation status would be a property. Please review the sentence.
Line 63: “brain diseases” you could consider rephrasing as “cerebral diseases.
Line 64: “good tool to suppress” you could consider replacing the word “good” with synonyms such as “appropriate tool”, in order to be more specific.
Line 66: “sensitivity of COVID-19” please consider referring to sensitivity to the virus, which is SARS-CoV-2, or susceptibility to the disease, which is COVID-19.
Line 68: “be important for suppressing a” could be grammatically correct as “important IN the suppression of”.
Line 71-74: “Accumulated studies have reported many aspects of high-density lipoprotein (HDL) in human growth, human disease, and the aging process under different environmental changes, such as dietary patterns and exposure to pathogens, smoking, and pollutants.” Please reference appropriate original articles on which this conclusion was based.
Line 78: Please spell out TC before introducing its abbreviation in text for the first time.
Line 87: “from 10s to 80s” the phrasing is ambiguous. It might refer to age, but I am unsure. Please revise.
Line 93: “…disease [17] and vascular dementia [18] in old age” please add the Oxford comma before “and” for better readability.
Line 99: “20s to 70s” Please consider replacing with 2nd to 7th decade of life or revise with an alternative phrasing.
Line 100-101: Appears in what is presented in the upcoming paragraph that women always have a higher level than men. Please consider if it is truly relevant to ask “when?”.
Line 167: Please spell out TG before introducing its abbreviation in text for the first time.
Line 174: Please spell out LpA-II before introducing its abbreviation in text for the first time.
Line 203: Please consider introducing the subfractions of HDL before mentioning them.
Line 210-214: Please rephrase. The comparisons are written in a slightly ambiguous manner.
Section 5.2: Please consider introducing spelled out forms before their abbreviations in text for the first time.
Line 251&256: Please omit full stops in the section/sub-section titles.
Line 253-255: Please give the citation for the quote. It would also be advisable to rephrase the quote.
Line 256-263: Please reconsider the usage of “good”, “better”, “best”. They are not specifically transmitting your point. Try alternatives that express your idea in a clearer manner, such as “healthier”.
Line 265: “The good quality of LDL contained 265 a small amount of oxidized species around 0.5±0.1” please omit “of”, as you are mentioning that “good quality LDL” contains the MDA.
Line 265-279: When comparing quantities, please keep them all in the same unit. Therefore, please convert all values to one unit (either nM or mg/dL).
Figure 1. Please cite the provenience of the figure. Please make sure you carefully explain what “good, bad, worst” mean in your view.
Line 297: “anti-infection” could be replaced with “anti-infective”.
Line 298: Please spell out HCV before introducing its abbreviation in text for the first time.
Line 300: “study of Chinese patients” please replace preposition with “on” instead of “of”.
Line 309: Please spell out SR-BI before introducing its abbreviation in text for the first time.
Line 341: Please spell out CE before introducing its abbreviation in text for the first time.
Line 342: Please keep the abbreviation of SR-BI consistent throughout text.
Line 348: Please spell out “day” in “54.9±23.3 g of alcohol/d”.
Line 352-356: Please spell out before introducing abbreviation in text for the first time. Please keep the abbreviations consistent throughout the text.
Line 358: Please spell out VLDL before introducing its abbreviation in text for the first time.
Line 373: Please spell out IL-1b, IL-6, and TNF-a before introducing its abbreviation in text for the first time.
Line 383: Please correct the typo in “a-s[yn”.
Line 394: Please spell out BBB before introducing its abbreviation in text for the first time.
Line 399: Consider dropping the comparative in “more synergistically”.
Line 411: Please add a comma (“…14 mg/dL, while SAA…”) for more facile readability.
Line 412: Please consider replacing “while” with an alternative, given its repetition in the sentence. Suggestion: “in contrast to the control group”.
Line 433: Please use the same comparative for the 2nd adjective: “healthiest”.
Line 434: Please replace “On the other hand” in this context with alternatives such as “however”.
Line 435: Please use the plural form “poluants”.
Line 440: Please spell out TEM before introducing its abbreviation in text for the first time.
Line 449: Please remove the dash in the section title.
Line 466: Please spell out RCT before introducing its abbreviation in text for the first time.
Line 477: Please spell out FC before introducing its abbreviation in text for the first time.
Line 480: Please consider introducing the subfractions of HDL before mentioning them.
Line 513: Please spell out apoE before introducing its abbreviation in text for the first time.
Line 523: Please spell out 4-HNE and iNOS before introducing its abbreviation in text for the first time.
Line 613-614: Please spell out PPAR before introducing its abbreviation in text for the first time.
Please make sure that re-used figures are given appropriate referencing and are obtained with consent of the authors publishing them.
I am very happy to have had the chance to review your insightful manuscript and I sincerely hope that the slight comments will be embraced as constructive.
Best regards,
One of your Reviewers
Reviewer 2 Report
The paper is well-written. However, the paragraph on the effects of nutritional supplementation on HDL is quite poor since discusses only 2 compounds. It should be updated or removed.
Author Response
The paper is well-written. However, the paragraph on the effects of nutritional supplementation on HDL is quite poor since discusses only 2 compounds. It should be updated or removed.
Dear reviewer,
Thank you for your helpful comments after careful reading of my manuscript.
I appreciate your valuable opinion to improve the manuscript
I revised manuscript in according to your comment as indicated in blue font.
I removed the sentences because the reference 46 was retracted.
“The combined supplementation of omega-3 (2g/d) and vitamin (vit) D in patients (n=26) with multiple sclerosis for 12 weeks resulted in an increase in serum HDL-C from 44.0±6.9 mg/dL (baseline) to 46.6±6.9 mg/dL (week 12), while the placebo group (n=27) showed a 43.3±6.8 mg/dL (baseline) to 44.1±6.8 mg/dL [46]. The differences in the mean outcome measures between treatment groups (b) were calculated to be 2.30 (p=0.009).”
The paragraph on the effects of nutritional supplementation on HDL has been updated as below.
“Meta-analysis of data from 33 randomized controlled trials revealed that dietary EPA supplementation on metabolic syndrome did not increase HDL-C (weighed mean difference, WMD=0.02 nmol/L), while DHA supplementation increased HDL-C (WMD=0.07 nmol/L) [46]. In contrast to inconsistent results from the combined supplementation, omega-3 (marine n-3) alone supplementation (1 g/d, n=12,933) for 5.3 years did not reduce the risk of major cardiovascular events than placebo without an increase of HDL-C compared to vit D3 (2000 IU/d) group (n=12,938) as a placebo [47].”
Reviewer 3 Report
- Some abbreviation as COVID-19, SARS-CoV-2, TC, and TG should be presented the full name at the first time in this manuscript.
- Line 512 & 532: Please only show the abbreviation.
Author Response
Dear reviewer,
Thank you very much for your helpful comments after careful reading of my manuscript.
I appreciate your valuable suggestion to improve the manuscript
All suggestions from the reviewer are reflected in this revision.
The revised sentences are indicated in blue font.
- Some abbreviation as COVID-19, SARS-CoV-2, TC, and TG should be presented the full name at the first time in this manuscript.
Agreed. The abbreviations are spelled out as reviewer’s suggestion.
coronavirus disease 2019 (COVID-19)
severe acute respiratory syndrome coronavirus 2 (SARS-CoV-2).
- Line 512 & 532: Please only show the abbreviation.
Agreed. The full spelling was removed to show only CVD and HDL-C, because they were appeared already.
Round 2
Reviewer 2 Report
Intriguing paper